# Kinetic competition during the transcription cycle results in stochastic RNA processing

Antoine Coulon[1†], Matthew L Ferguson[2†‡], Valeria de Turris[3], Murali Palangat[2], Carson C Chow[1], Daniel R Larson[2*]

[1]Laboratory of Biological Modeling, National Institute of Diabetes and Digestive and Kidney Diseases, National Institutes of Health, Bethesda, United States; [2]Laboratory of Receptor Biology and Gene Expression, Center for Cancer Research, National Cancer Institute, National Institutes of Health, Bethesda, United States; [3]Center for Life Nanoscience, Istituto Italiano di Tecnologia, Rome, Italy

**Abstract** Synthesis of mRNA in eukaryotes involves the coordinated action of many enzymatic processes, including initiation, elongation, splicing, and cleavage. Kinetic competition between these processes has been proposed to determine RNA fate, yet such coupling has never been observed in vivo on single transcripts. In this study, we use dual-color single-molecule RNA imaging in living human cells to construct a complete kinetic profile of transcription and splicing of the β-globin gene. We find that kinetic competition results in multiple competing pathways for pre-mRNA splicing. Splicing of the terminal intron occurs stochastically both before and after transcript release, indicating there is not a strict quality control checkpoint. The majority of pre-mRNAs are spliced after release, while diffusing away from the site of transcription. A single missense point mutation (S34F) in the essential splicing factor U2AF1 which occurs in human cancers perturbs this kinetic balance and defers splicing to occur entirely post-release.

*For correspondence: dan.
larson@nih.gov

†These authors contributed
equally to this work

Present address: ‡Department
of Physics, Boise State University,
Boise, United States

**Competing interests:** The
authors declare that no
competing interests exist.

**Reviewing editor**: Douglas L
Black, Howard Hughes Medical
Institute, University of California,
Los Angeles, United States

## Introduction

Co-transcriptional processing of nascent pre-mRNA is a central mechanism for gene regulation in eukaryotes and requires temporal coordination between transcription initiation, elongation, splicing, and cleavage. Each of these processes is carried out by megadalton macromolecular complexes acting at a single genetic locus, and kinetic competition between these processes has been proposed to determine RNA fate (*Bentley, 2014*). Genome-wide studies across organisms indicate heterogeneous distributions of both RNA polymerase and nascent RNA along the gene, suggesting that kinetic checkpoints exist throughout the gene, including at promoter-proximal sites, translation start sites, intron–exon boundaries, and at the 3′ end of genes (*Core et al., 2008*; *Nechaev et al., 2010*; *Churchman and Weissman, 2011*; *Hah et al., 2011*; *Larson et al., 2014*). However, population studies reflect the balance of kinetic rates and are unable to resolve the multiple competing processes occurring at a single gene. Moreover, genome-wide measurements lack the time-resolution which might provide mechanistic clues about the underlying enzymatic processes.

The hypothesis of kinetic competition is that a fast process will out-compete a process which may in fact be more energetically preferred. Kinetic competition during the transcription cycle has been shown to influence splice site selection during alternative splicing (*de la Mata et al., 2003*), recruitment of factors which release promoter-proximal pausing (*Li et al., 2013*), and even RNAi-mediated genome defense (*Dumesic et al., 2013*). Since these processes occur within the dynamic milieu of the nucleus, the stochastic interactions between macromolecules may result in a range of possible outcomes

**eLife digest** To make a protein, part of a DNA sequence is copied to make a messenger RNA (or mRNA) molecule in a process known as transcription. The enzyme that builds an mRNA molecule first binds to a start point on a DNA strand, and then uses the DNA sequence to build a 'pre-mRNA' molecule until a stop signal is reached.

To make the final mRNA molecule, sections called introns are removed from the pre-mRNA molecules, and the parts left behind—known as exons—are then joined together. This process is called splicing. However, it is not fully understood how the splicing process is coordinated with the other stages of transcription. For example, does splicing occur after the pre-mRNA molecule is completed or while it is still being built? And what controls the order in which these processes occur?

One theory about how the different mRNA-making processes are coordinated is called kinetic competition. This theory states that the fastest process is the most likely to occur, even if the other processes use less energy and so might be expected to be preferred. Alternatively, the different steps may be started and stopped by 'checkpoints' that cause the different processes to follow on from each other in a set order.

Coulon et al. used fluorescence microscopy to investigate how mRNA molecules are made during the transcription of a human gene that makes a hemoglobin protein. To make the RNA visible, two different fluorescent markers were introduced into the pre-mRNA that cause different regions of the mRNA to glow in different colors. Coulon et al. made the introns fluoresce red and the exons glow green. Unspliced pre-mRNA molecules contain both introns and exons and so fluoresce in both colors, whereas spliced mRNA molecules contain only exons and so only glow with a green color.

By looking at both the red and green fluorescence signals at the same time, Coulon et al. could see when an intron was spliced out of the pre-mRNA. This revealed that in normal cells, splicing can occur either before or after the RNA is released from where it is transcribed. Thus, splicing and transcription does not follow a set pattern, suggesting that checkpoints do not control the sequence of events. Instead, the fact that a spliced mRNA molecule can be formed in different ways suggests kinetic competition controls the process.

In some cancer cells, there are defects in the cellular machinery that controls splicing. When looking at cells with such a defect, Coulon et al. found that splicing only occurred after transcription was completed. This study thus provides insight into the complex workings of mRNA synthesis and establishes a blueprint for understanding how splicing is impaired in diseases such as cancer.

for the nascent RNA. Stochastic RNA synthesis—the phenomenon whereby the inherently stochastic nature of bio-molecular encounters and reactions leads to a non-deterministic production of transcripts—has been directly visualized in multiple organisms (*Golding et al., 2005*; *Chubb et al., 2006*; *Yunger et al., 2010*; *Larson et al., 2011*, *2013*). Yet, stochastic RNA processing—the possibility that stochastic bio-molecular reactions might lead to non-deterministic pathways/outcomes in the making and maturation of an RNA—has never been directly observed, and the potential consequences for gene regulation are largely unexplored. Alternatively, regulatory checkpoints have been proposed which safeguard against such stochastic RNA processing events, providing a level of quality control. For example, the model of exon definition requires that splicing of the terminal intron relies on synergy between 3′ end formation, nascent RNA cleavage, and intron excision (*Berget et al., 1977*; *Niwa et al., 1990*). Similarly, multiple studies indicate an increased density of nascent RNA present at the 3′ end of genes or in the chromatin-bound fraction, suggesting that nascent RNA is retained at the site of transcription to ensure correct processing (*Glover-Cutter et al., 2008*; *Brody et al., 2011*; *Carrillo Oesterreich et al., 2010*; *Bhatt et al., 2012*). In both the competition model and the checkpoint model, kinetics plays a prominent role, but in the latter case, the cell has developed additional safeguard mechanisms.

In this article, we use an in vivo single-molecule RNA imaging approach to directly measure kinetic coupling between transcription and splicing of a human β-globin reporter gene. The approach is based on simultaneous dual-color imaging of both the intron and exon of the same pre-mRNA using both PP7 and MS2 stem loops (*Bertrand et al., 1998*; *Chao et al., 2008*). We find that kinetic

competition results in multiple competing pathways for pre-mRNA splicing. Splicing of the terminal intron occurs stochastically both before and after transcript release, indicating there is not a strict quality control checkpoint. Post-release splicing occurs from freely diffusing transcripts in the nucleus and is an order of magnitude faster than splicing at the site of transcription. A single missense mutation (Ser34Phe) in the zinc finger domain of the conserved splicing factor U2AF1 which is recurrent in multiple cancers (*Yoshida et al., 2011*; *Graubert et al., 2012*; *Waterfall et al., 2014*) changes the balance, making all splicing post-release. This same effect can also be observed on the endogenous, un-modified fragile X mental retardation mRNA (*FXR1*). Our results show that kinetic competition governs the stochastic balance between multiple competing pathways for RNA synthesis and processing and that this balance is perturbed by oncogenic mutations.

## Results

### Real-time visualization of transcription and splicing at the single-molecule level in living cells

To visualize transcription, splicing, and release of single transcripts in living cells, we used time-lapse fluorescence microscopy of multiply labeled RNA (*Bertrand et al., 1998*). We stably integrated into U2-OS cells, a human β-globin reporter with a DNA cassette that encodes for 24X PP7 RNA hairpins in the second intron (*Chao et al., 2008*) and a 24X MS2 hairpin cassette in the 3′ UTR (*Boireau et al., 2007*) (*Figure 1A*). The constitutively expressed PP7-coat protein tagged with mCherry (PCP-mCherry) and MS2-coat protein tagged with GFP (MCP-GFP) bind with high affinity (the on rate for MCP is 0.54 μM$^{-1}$s$^{-1}$. At a nuclear concentration of 1 μM, the average time for the MCP to bind a completed stem loop is 1.85 s) (*Buenrostro et al., 2014*) to the RNA stem loops as homodimers, tagging each

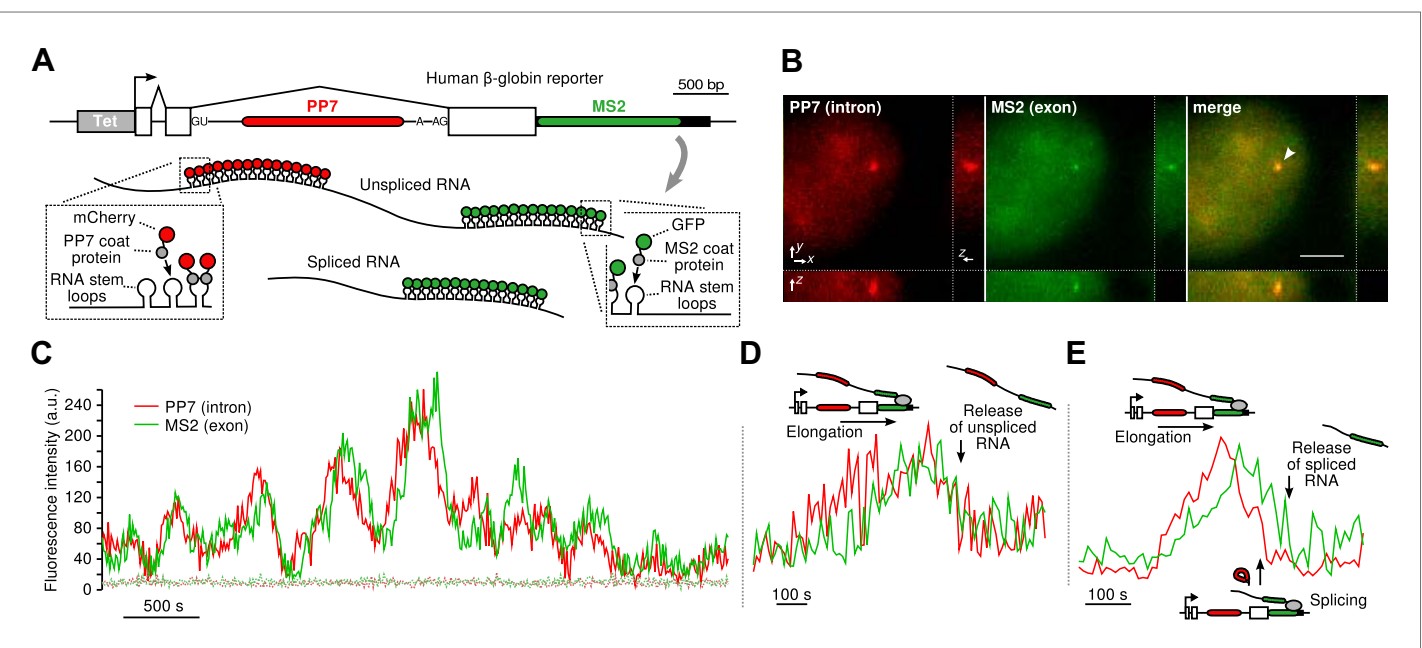

**Figure 1**. Real-time measurement of transcription and splicing in living cells. (**A**) Schematic of the human β-globin report gene construct. Reporter splicing efficiency >95% by qRT-PCR (*Figure 1—figure supplement 1C*). (**B**) 3D images of diffraction-limited spot in both channels corresponding to the transcription site (TS, arrow). Bar: 4 μm. (**C**) Fluorescence fluctuations recorded at the TS reflect stochastic transcriptional events. Dotted lines are background traces recorded in the nucleus, 8 μm away from the TS. (**D** and **E**) Examples of pre- and post-release splicing observed when the intron (red signal) disappears simultaneously with (**D**) or before (**E**) the exon (green signal).

The following figure supplements are available for figure 1:

**Figure supplement 1**. Human β-globin reporter gene.

**Figure supplement 2**. Integration site of the β-globin reporter and copy number analysis.

cassette with 48 fluorophores of a single color, resulting in orthogonal labeling of two different parts of the nascent transcript (*Hocine et al., 2013*; *Martin et al., 2013*; *Buenrostro et al., 2014*). Since the PP7 cassette is intronic, unspliced RNAs appear in both colors, while spliced RNAs are only visible in green. Time-lapse imaging of cells in 3D reveals a temporally fluctuating diffraction-limited spot, co-localizing in both colors, that corresponds to the transcription site (TS) where nascent transcripts are synthesized (*Figure 1B–C*, *Videos 1* and *Videos 2*). We observed mature mRNA (exon only) diffusing in both the nucleus and the cytoplasm (*Video 3*), and we verified expression of the protein product, indicating the message is spliced and translated correctly (*Figure 1—figure supplement 1B–C*).

By simultaneously observing the fluorescence intensity of the intron and the exon of a single nascent transcript, one can determine when the intron is excised from the pre-mRNA. We find that both pre- and post-release splicing are visible as single events at the same gene over time (*Figure 1C*). In the case of post-release splicing, the intronic fluorescence appears, followed by the exon fluorescence, followed by a coincidental drop in both colors reflecting the release of an unspliced RNA (*Figure 1D* and *Figure 1—figure supplement 1E*). For pre-release splicing, there is a delay between the drop-off of the red and green signals, indicating intron removal before release (*Figure 1E*). Diffusion of the pre-mRNA away from the TS is rapid, which accounts for the precipitous drop of the signal in the time trace (*Figure 1C*).

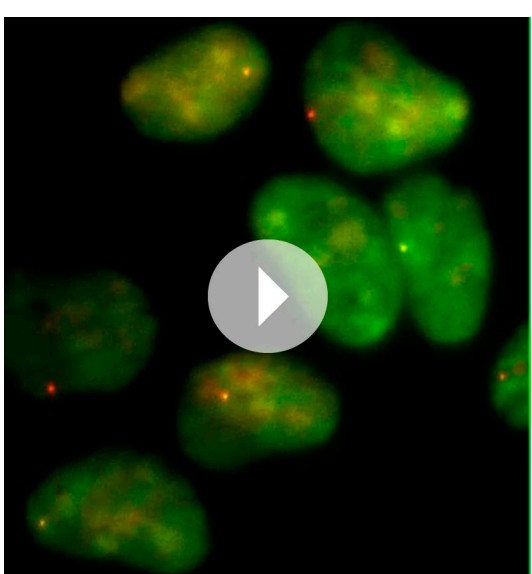

**Video 1**. Time-fluctuating transcription sites. Cells show a diffraction-limited fluorescent spot colocalizing in both colors (red: intron, green: exon), corresponding to the transcription site of the reporter gene. The fluorescence intensity of each site fluctuates over time as nascent transcripts are synthesized, spliced, and released from the transcription site. Large orange shapes in nuclei are nucleoli (*Ferguson and Larson, 2013*).

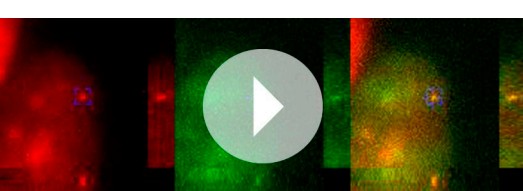

**Video 2**. Tracking of a transcription site in 4D. The video shows, for the intron and exon signals (left and center panels), the maximum intensity projected image from the top (square image) and from the sides (rectangle images), revealing the transcription site (TS) in three dimensions (3D) and over time (4D). Image analysis is used to track the TS over time in both colors. The blue box and cross indicate the location of the TS as found by the tracking algorithm. The right panel is the merge of both signals.

## Fluctuation analysis reveals the stochastic synthesis and processing kinetics of single transcripts

Most of the time, multiple nascent RNA is present at the TS, necessitating a general analysis method for extracting kinetic information from the time traces (*Larson et al., 2011*). We developed a dual-color fluctuation correlation analysis approach for analyzing the complete transcription cycle, resulting in two temporal auto-correlation functions and a single temporal cross-correlation function (*Figure 2A*; See 'Materials and methods'). These functions measure, over many traces, how a fluctuation in one fluorescence channel is statistically correlated with a fluctuation in either the same or the other channel after a given time delay.

The correlation functions for β-globin (*Figure 2B*, *Figure 2—figure supplement 1A–B*) reveal the kinetic features hidden in the fluorescence time traces and encapsulate the stochastic kinetics of single-transcript synthesis (*Figure 2—figure supplement 2*, *Supplementary file 1*—§2). At short time delays (<40 s), the cross-correlation reflects the order of splicing and release events (*Figure 2B–C* and *Supplementary file 1*—§2). If splicing never occurs before transcript release, the intron-to-exon cross-correlation (blue) starts horizontally at positive delays. On the other hand, if splicing always occurs before release, this function starts with a positive slope in alignment with its negative counterpart (magenta). In the case

**Video 3**. Spliced RNAs diffusing in the nucleus and the cytoplasm. Cells are imaged here with a high laser power and a short exposure time so that diffusion of single RNAs can be appreciated. It reveals a population of transcripts diffusing in both the nucleus and the cytoplasm, as evidenced by fast fluctuations observed in the exon signal (right panel). These transcripts are, for the most part, already spliced since the intron signal (center panel) does not show the same fluctuations. In these imaging conditions, unspliced transcripts are only visible at the transcription site (bright spot in the nucleus colocalizing in both color; see merge in left panel).

where splicing is stochastic and both outcomes may occur, their relative probability of occurrence is given by the change of slope of the cross-correlation at 0 delay (*Figure 2C*, *Figure 2— figure supplements 2D and 3*). The experimental cross-correlation determined from ~2000 individual β-globin transcripts indicates that splicing occurs before release for a fraction of transcripts (13 ± 5%, *Figure 2B* inset). We note that this change in slope at short time delay *only* denotes the relative order of splicing *vs* release but says nothing about the kinetics of these two processes. In summary, these data demonstrate that splicing and release are not firmly constrained to occur in a specific order (p < 0.003).

At longer delays (>40 s), other features of the transcription cycle are visible. For example, the delay at which the intron-to-exon cross-correlation (*Figure 2B*, blue circles) starts decreasing (~60 s) corresponds to the elongation time between the two cassettes (2573 bases apart), resulting in an elongation rate of ~2.6 kb/min. Finally, the decay at time-scales > 100 s relates to the dwell time of transcripts at the TS (*Figure 2—figure supplement 2A,B*). Specifically, the long decays observed in all correlation functions indicates that RNA is not immediately released after transcription of the poly(A) site. Rather, the transcript remains at the TS for a duration which reflects either a pause at/near the poly(A) site, transcription past the termination site, or a post-cleavage retention of the transcript within the diffraction-limited spot (*Hofer and Darnell, 1981*; *Glover-Cutter et al., 2008*; *Brody et al., 2011*; *Carrillo Oesterreich et al., 2010*; *Bhatt et al., 2012*).

To confirm our assignment of transcription cycle events to features in the correlation curve, we treated cells with drugs known to affect different aspects of RNA synthesis. The splicing inhibitor spliceostatin A (SSA) (*Kaida et al., 2007*) abolished splicing at the TS as evidenced by the disappearance of the rise in the intron-to-exon correlation function (*Figure 2D*). Treatment with camptothecin (CPT, a topoisomerase I inhibitor known to slow down elongation [*Singh and Padgett, 2009*]) resulted in a marked shift of the decreasing part of the intron-to-exon cross-correlation to longer delays (*Figure 2E*), which is the expected manifestation of slower elongation (*Figure 2—figure supplement 2A–B*, *Supplementary file 1—§2*).

As an additional control, when shuffling channels between traces or when using time traces recorded away from the TS, the correlation functions are flat (*Figure 2—figure supplement 1G–H*), supporting the fact that the correlation functions shown on *Figure 2B* reflect the molecular events happening at the TS.

Finally, we emphasize that an essential advantage of this approach is that correlation functions reveal single-transcript kinetics even from signals where multiple transcripts are present at any given time (*Supplementary file 1—§1*). As illustrated in *Figure 3*, a single transcription and splicing event results in correlation functions with a peak near zero delay, reflecting the *intra*-transcript kinetics (*Figure 3A*). If three transcripts are present, additional peaks appear at non-zero delay, due to *inter*-transcript kinetics but all the correlations resulting from *intra*-transcript kinetics accumulate around 0 (*Figure 3B*). After averaging over many transcription and splicing events, *inter*-transcript correlations disappear, leaving only a central peak which reflects the kinetics of single transcripts (*Figure 3C*). See also *Video 4* and *Figure 3—figure supplement 1*.

## Kinetic competition between splicing and transcript retention determines the balance between pre- and post-release splicing

The preceding conclusions are general and make no reference to a specific model. To gain further insight, we developed mathematical models which relate the shape of the correlation functions to the timing of the underlying molecular processes (see 'Materials and methods'). We generated five different mechanistic schemes: (I) purely post-release splicing, (II) independence between splicing and elongation/release, (III) polymerase pausing at the 3' splice site (ss) until splicing is complete,

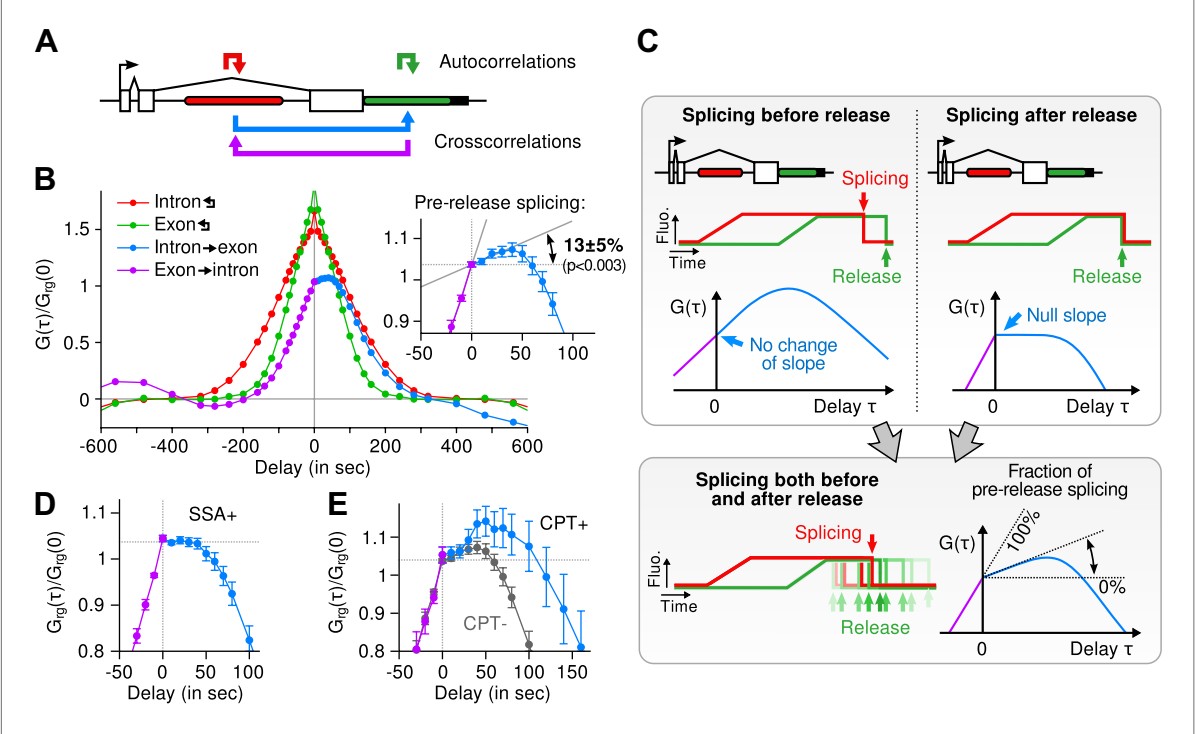

**Figure 2**. Transcription and splicing kinetics are revealed by fluctuation analysis of dual-color fluorescence intensity time traces. (**A**) Auto- and cross-correlation functions quantify statistically correlated fluctuations occurring at different time delays, respectively within the same or between two signals. (**B**) Correlation functions ($G(\tau)$) of experimental time traces (N = 21). Auto-correlations (red and green curves) are symmetrical by construction. Cross-correlations (blue and magenta curves) are two halves of a single continuous curve. Inset: short-delay behavior of the cross-correlation reveals that 13 ± 5% of the RNAs are spliced pre-release (p-value: pre-release fraction ≠ 0% and 100%; z-test). (**C**) Schematic representing stochastic pre- and post-release splicing. Purely pre-release splicing imposes the cross-correlation to have the same rising slope on both sides of the y-axis, while purely post-release makes the intron-to-exon cross-correlation (blue curve, positive delay) start as a plateau. The change of slope at τ = 0 delay is indicative of the fraction of splicing events occurring before release. (**D**) Spliceostatin A abolishes pre-release splicing. (**E**) Camptothecin delays the decay of the intron-to-exon cross-correlation and increases the pre-release fraction. All correlation functions are normalized by the value of the cross-correlation at 0 delay ($G_{rg}(0)$). Error: SEM (bootstrap). Control correlation functions are shown in *Figure 2—figure supplement 1G–H*.

The following figure supplements are available for figure 2:

**Figure supplement 1**. Fluorescence time traces and correlation functions.

**Figure supplement 2**. Geometry of the correlation functions.

**Figure supplement 3**. Estimation of the fraction of pre-release splicing from slopes in the crosscorrelations.

**Figure supplement 4**. Mechanistic schemes.

**Figure supplement 5**. Model comparison using a Bayesian Information Criterion (BIC).

**Figure supplement 6**. Counting transcripts at the transcription site.

(IV) splicing only during 3′ end retention of the transcript, and (V) release only after splicing is complete (*Figure 2—figure supplement 4* and 'Materials and methods'). For each one of these general schemes, different time distributions were tested for elongation, splicing, and release (*Supplementary file 2*). Since by construction, the intron-to-exon cross-correlation at 0 delay is necessarily null in scheme III and have a null slope in scheme I, these two schemes can be ruled out (See *Figure 2—figure supplement 5A,D* for fits). The three other schemes were better at fitting the correlation curves but the best model is one from scheme II, that is where splicing is independent of elongation and transcript release

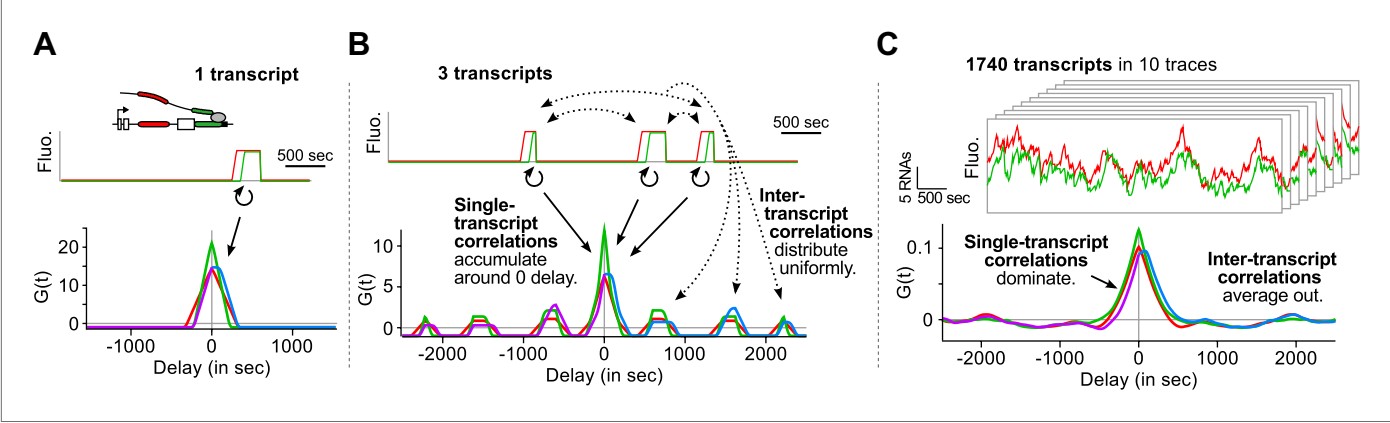

**Figure 3**. Correlation functions reflect single-transcript kinetics. (**A**) A dual-color time trace with a single transcription event yields correlation functions with features around 0 delay and flat elsewhere. (**B**) When several transcription events are present in a time trace, the correlation coming from each individual RNA accumulates around 0 delay, while all the correlation between pairs of RNAs distributes uniformly on the delay axis. (**C**) When there are many transcription events per time trace and/or many traces are used to produce an average correlation function, the correlation from single transcripts dominates and that from pairs of transcripts averages out. The resulting correlation functions hence reflect single-transcript kinetics. Time traces shown are simulations where the statistics of transcript kinetics are similar to those we measured by live cell imaging. Traces in (**C**) have the same duration and number of transcripts as estimated in experimental data (e.g., **Figure 1C**). See **Video 4** for an animation of how the correlation functions converge as the number of transcripts increases.

The following figure supplement is available for figure 3:

**Figure supplement 1**. Correlation functions with several gene copies at the TS reflect single-transcript kinetics.

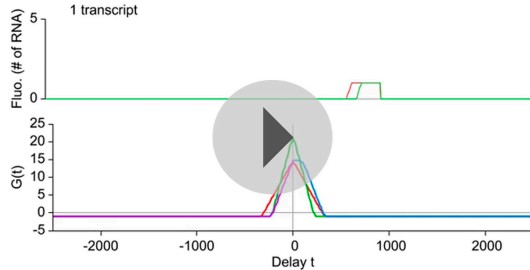

**Video 4**. Correlation functions reveal single transcript kinetics. This video shows the convergence of the correlation functions for increasing number of transcripts in a time trace. See also **Figure 3**.

(See **Figure 2—figure supplement 5** and discussion on **Model comparison** in 'Materials and methods'). In this 3-parameter model (**Table 1**), splicing occurs a fixed amount of time after the 3'ss has been transcribed, and transcript release involves a stochastic delay after the poly(A) site is reached. No pause at the 3' ss was needed to fit the data. This observation does not rule out pausing at these sites but suggests that such a pause would be much shorter than the other timescales observed. Notably, our data are fit better with a model having a fixed time for intron removal rather than a stochastic (exponential) one, arguing for several sequential kinetic steps (**Aitken et al., 2011**; **Schmidt et al., 2011**). In total, the β-globin-terminal intron splicing time was 267 ± 9 s after the polymerase passes the 3' ss. This measurement of splicing time is consistent with previous estimates either in vivo on cell populations (**Singh and Padgett, 2009**) or in vitro at the single-molecule level (**Hoskins et al., 2011**), suggesting that PP7 stem loops do not perturb splicing kinetics of this intron, contrary to MS2 stem loops (**Aitken et al., 2011**; **Schmidt et al., 2011**). As an independent validation of our modeling results, we counted the number of red and green RNAs at transcription sites using a normalized ratiometric approach (**Zenklusen et al., 2008**) (See 'Materials and methods' and **Figure 2—figure supplement 6**). The average red-to-green ratio of 1.41 is indistinguishable from the expected 1.4 value predicted by our modeling analysis of the correlation functions.

Importantly, both SSA and CPT drug treatments only affected a single parameter (**Table 1**), arguing that splicing is kinetically independent of elongation/termination. Based on our measurements, splicing occurring at the TS is rarely completed during elongation but rather during a pause at the 3' end of the gene. Because transcript release is stochastic, the 3' end dwell time can be shorter or longer than

the necessary time to remove the intron, resulting in splicing that can occur either before or after release.

## Splicing happens 10-fold faster on freely diffusing transcripts than on chromatin

Since the majority of β-globin pre-mRNA is released from the TS before splicing of the terminal intron, we addressed the question of where and when this splicing takes place. We observed a mobile population of unspliced pre-mRNA (co-localized intron/exon) diffusing in the immediate vicinity of the TS (*Figure 4A–B*, *Video 5*). In contrast, spliced mRNA (exon only) could be observed diffusing throughout the nucleus. A small population of red-only particles was also recorded, which could be due either to the false-discovery rate of the segmentation algorithm or the presence of free lariats (See 'Materials and methods'). The radial distribution of mRNA and pre-mRNA indicated an enrichment for unspliced transcripts within 2.4 ± 0.1 μm of the TS (*Figure 4C*, *Figure 4—figure supplement 1*), meaning that splicing occurs faster than diffusion throughout the nucleus. This enrichment disappears upon treatment with splicing inhibitor SSA, in which case most of the transcripts are unspliced and dispersed throughout the nucleus (*Figure 4C*, *Video 6*). From the measured diffusion coefficient (D = 0.12 μm$^2$/s, *Figure 4—figure supplement 2*; See 'Materials and methods'), we calculated that post-release splicing takes place on average 13 ± 1 s after departure from chromatin. This time is much shorter than the expected 137 s it would take if pre- and post-release splicing kinetics were identical (calculated from *Table 1*). From this observation, it is tempting to speculate that transcripts are released only after they have passed a particular rate-limiting step in spliceosome assembly, explaining why the catalytic step occurs very soon after release. However, this interpretation is inconsistent with the fact that 3′end retention time is not affected by SSA treatment which impairs binding of U2 and hence affects recruitment of all the snRNPs except U1 (*Corrionero et al., 2011*). Note also that we cannot formally exclude the possibility that co-localized intron/exon particles are actually excised introns still in complex with transcripts. In summary, although introns can be retained for over 4 min on chromatin, once the transcript is released, splicing is 10-fold faster for freely diffusing transcripts. In most cases, the nascent intron is retained until transcription reaches the 3′ end of the gene and then removed either before or shortly after release of the transcript (*Figure 5*).

## Recurrent cancer-associated mutation in splicing factor U2AF1 delays splicing and makes it entirely post-release

Because the balance of kinetic competition determines where and when introns are excised from the pre-mRNA, we then sought to determine whether *trans*-acting factors regulated splicing by perturbing this balance. Deep sequencing studies in myelodysplastic syndrome, chronic lymphocytic leukemia, acute myeloid leukemia (AML), breast cancer, lung adenocarcinoma, and hairy cell leukemia have all revealed the existence of mutated factors involved in 3′ ss recognition (*Yoshida et al., 2011*; *Graubert et al., 2012*; *Brooks et al., 2014*; *Waterfall et al., 2014*; *TCGA, 2012*). One recurrent change is a heterozygous point mutation in U2 auxiliary factor 1 (U2AF1), which is an essential factor for recognition of the AG dinucleotide consensus motif (*Figure 1A*). The serine-to-phenylalanine (S34F) missense mutation in the zinc finger domain results in disparate changes in alternative splicing

**Table 1.** Kinetics of transcription and splicing under different experimental conditions

| | Elongation rate (kb/min) | Mean 3' end dwell time (sec) | Splicing time (sec) | Pre-release fraction (%) |
|---|---|---|---|---|
| Control | 2.60 ± 0.16 | 116.1 ± 5.8 | 267 ± 9 | 15.9 ± 3.2 |
| SSA+ | 2.41 ± 0.26 | 126.7 ± 5.7 | 485 ± 62 ** | 3.5 ± 2.4 ** |
| CPT+ | 1.44 ± 0.09 ** | 111.0 ± 10.3 | 251 ± 10 | 24.9 ± 6.9 |
| U2AF1 (wt) | 2.24 ± 0.27 | 120.7 ± 4.9 | 280 ± 8 | 16.6 ± 3.3 |
| U2AF1-S34F | 2.64 ± 0.11 | 166.0 ± 7.0 ** | 694 ± 176 * | 2.1 ± 2.6 ** |

The table shows result of fits with model II.4 ('Materials and methods' and **Supplementary file 2**). Pre-release fraction is deduced from the 3 other parameters. Errors are propagated SEM from correlation functions. * p-value<0.05, ** p-value<0.005 (two-sided z-test vs control).

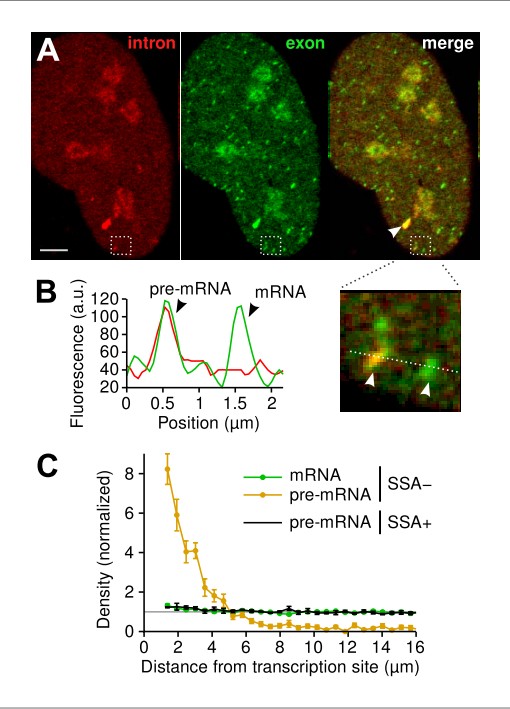

**Figure 4**. Visualization of splicing occurring after release from chromatin. (**A**) Individual frames from live-cell confocal imaging showing intron (red dots), exon (green dots), and the merged image. White arrow: TS. Bar: 4 μm. (**B**) Fluorescence intensity profile along the line in the inset shows co-localized intron/exon (unspliced pre-mRNA) and exon only (spliced mRNA). (**C**) Radial distributions of mRNA (green) and pre-mRNA (orange), as well as pre-mRNA under SSA treatment (black) are shown as a function of distance from the TS. Density distributions are normalized by the distribution of random (uniform) positions within the nucleus (see 'Materials and methods'). Error: SEM (bootstrap over 9 cells).

The following figure supplements are available for figure 4:

**Figure supplement 1**. Localization of single RNAs diffusing in the nucleus.

**Figure supplement 2**. Measure of RNA diffusion in the nucleus by RICS.

patterns including exon skipping, exon inclusion, and alternative 3' ss selection (***Brooks et al., 2014***). Although somatic genetics clearly indicate the importance of this mutation (***Waterfall et al., 2014***), there is no functional or mechanistic understanding of how U2AF1-S34F works at the molecular level.

We performed time-lapse imaging on cells expressing moderate levels of either wild-type U2AF1 or U2AF1-S34F, both fused to a cerulean fluorescent protein. We note that this experimental condition recapitulates the in situ case, because the mutant U2AF1 is present against the background of at least one copy of the wild-type allele. Correlation functions revealed that U2AF1-S34F completely abolishes pre-release splicing (***Figure 6A***, horizontal slope) and prolongs transcript 3' end dwell time (***Table 1***). Post-release imaging of transcripts showed a local enrichment for unspliced pre-mRNA near the TS, but with a greater spatial extent in the case of U2AF1-S34F (***Figure 6B–C***, ***Video 7***). Thus, all transcripts are post-transcriptionally spliced, albeit at a slower rate (27 ± 3 s; ***Figure 6D***, ***Figure 4—figure supplement 1***). Splicing efficiency, poly(A) length, and 3' UTR length were unchanged (***Figure 1—figure supplement 1C–D***). In summary, these data suggest that the mutant delays splicing to post-release, slows splicing from freely diffusing transcripts, but has no detectable effect on splicing efficiency.

We then confirmed this kinetic effect on the endogenous FXR1 mRNA, which was shown to be alternatively spliced in the presence of U2AF1-S34F in both AML and lung adenocarcinoma (***Brooks et al., 2014***). Using single-molecule FISH (***Femino et al., 1998***) (***Figure 6E***), we examined the spatial distribution of intron and exon in fixed cells transfected with U2AF1 or U2AF1-S34F. As was the case for the β-globin reporter, we observed a population of unspliced FXR1 pre-mRNA in the vicinity of an active TS, indicating that at least some fraction of FXR1 transcripts are released before splicing (***Figure 6F***). We then performed the same spatial analysis as above, except on fixed cells instead of live cells. Expression of the U2AF1-S34F mutant resulted both in an increase in the level of unspliced pre-mRNA in the nucleus (16.0 ± 0.4% compared to 6.0 ± 0.3%, ***Figure 6F***) and the radial distance from the TS (1.8 ± 0.5 μm compared to 0.5 ± 0.2 μm, ***Figure 6G***). Taken together, both static and dynamic measurements on a reporter β-globin transcript and the endogenous FXR1 transcript suggest that the S34F mutation in U2AF1 acts in a dominant negative fashion to postpone splicing until after release and cause slower splicing from diffusing transcripts.

## Discussion

The picture that emerges from this study is one in which the β-globin-terminal intron can be spliced during multiple steps of the transcription cycle (***Figure 5***). A minority of transcripts are spliced while

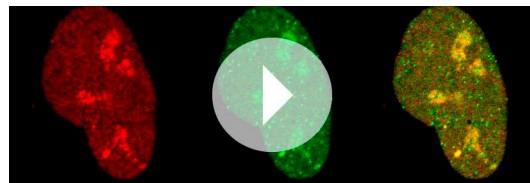

**Video 5**. Single-RNA imaging reveals a transient population of unspliced transcript diffusing away from the transcription site. Using high-power confocal laser scanning microscopy, we were able to observe single transcripts with a better temporal resolution than with widefield imaging (**Video 3**). The video shows a single cell with an active transcription site (TS, bright spot visible in both signals) and diffusing RNA particles (left: intron, center: exon, right: merge). Although most of the RNAs diffusing in the nucleus are spliced (visible only in the exon signal), few unspliced RNAs (visible in both colors) are detectable in the vicinity of the TS as they diffuse away. Spatial distribution and diffusion analyses revealed that this population is very transient (**Figure 4C** and **Figure 4—figure supplements 1 and 2**). Large shapes in the nucleus are nucleoli (**Ferguson and Larson, 2013**).

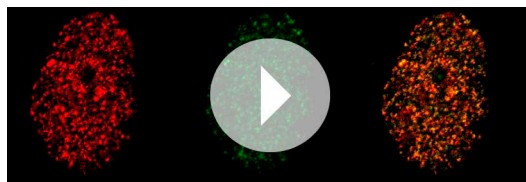

**Video 6**. Single-RNA imaging with splicing inhibitor SSA. Imaging conditions are identical as in **Video 5**, but cells are treated with splicing inhibitor spliceostatin A (SSA). RNAs diffusing in the nucleus are now visible in both color, indicating that they are unspliced.

retained at the 3′ end of the gene, but the dominant pathway is the one in which splicing is contemporaneous with release or occurs shortly thereafter. These results are consistent with several studies which suggested that intron removal is enhanced upon cleavage (**Baurén et al., 1998**; **Bird et al., 2005**). In fact, we find that splicing is 10-fold faster on freely diffusing transcripts than on chromatin-bound transcripts. Our data are consistent with a model where commitment to splicing of the terminal β-globin intron occurs co-transcriptionally but suggest there may be a high energy barrier to completion of intron removal while transcripts are still tethered to chromatin, possibly due to steric constraints. Thus, post-release splicing may be the energetically favored process, but there is a long kinetic window in which the less-favorable pre-release intron removal can occur.

The time required to remove an intron therefore becomes a central parameter in our understanding of RNA processing, with implications for both constitutive and alternative splicing (**Bentley, 2014**). This fundamental kinetic quantity has been elusive. Population-based measurements suggest a splicing time for several endogenous, un-modified introns of 5–10 min (**Singh and Padgett, 2009**). Bulk in vitro measurements on β-globin indicate a 40- to 50-min timescale. A single-molecule in vitro study on yeast transcripts measured ~10-min splicing time. On the opposite end of the spectrum, previous live-cell imaging approaches relying on both direct and indirect measurements indicate splicing times in the order of ~30 s (**Huranová et al., 2010**; **Martin et al., 2013**). One of the primary experimental advances in our study is the ability to observe the transcription and splicing process with a time resolution

that spans three orders of magnitude. Since the time to splice is a distributed quantity, and splicing times vary from transcript to transcript, the variation in previously reported splicing rates may be strongly influenced by the temporal dynamic range of the method. Biologically, this variability in splicing rate may provide regulatory potential, as we discuss below.

One potential criticism of the live-cell imaging approach is that the stem-loops and the coat protein may perturb kinetics. Several arguments stand against this view. First, our splicing kinetics are consistent with both population measurements (**Singh and Padgett, 2009**) and in vitro single-molecule measurements (**Hoskins et al., 2011**). Second, the splicing efficiency of the reporter is high (**Figure 1—figure supplement 1C**), suggesting there are no dead-end intermediates. Third, endogenous metazoan RNAs are decorated with RNA-binding proteins from their inception (**Castello et al., 2012**), suggesting that the spliceosome is well-equipped to handle bulky messages. Finally, since we can never exclude the possibility that a synthetic reporter may be missing features found in an endogenous gene, we have also recapitulated the results on the un-modified endogenous FXR1 message.

Importantly, our single-molecule study reveals the existence of multiple pathways, indicating the absence of a strict checkpoint for intron removal (**Bird et al., 2005**; **Alexander et al., 2010**; **Brody et al., 2011**; **Carrillo Oesterreich et al., 2010**; **Bhatt et al., 2012**; **Pandya-Jones et al., 2013**). The presence of a delay at the 3′ end may be interpreted as a checkpoint mechanism to ensure that splicing takes place before transcript release, but there are several reasons to reject this interpretation of

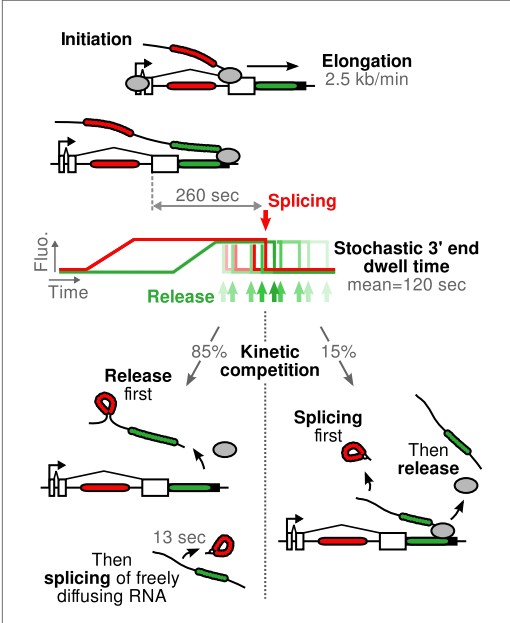

**Figure 5**. Schematic of β-globin transcription cycle kinetics. Transcript synthesis and processing can occur through different pathways, the choice of which is governed by a kinetic competition between transcription and splicing. After transcription of the 3′ splice site, intron removal takes about 260 s and elongation until the end of the gene, about 55 s. Hence, splicing does not occur during elongation. The transcript is retained at the 3′end of the gene for a stochastic amount of time that can be shorter or longer than the remaining time to excise the intron. This results in two possible outcomes: either an unspliced pre-mRNA is released and then spliced very rapidly or splicing occurs while the transcript is retained on chromatin before being released.

our data. First, none of the models which assume dependence between splicing and elongation/release fit better than one that assumes independence. Second, both splicing and elongation inhibition experiments support the view of the two processes being kinetically independent. Abolishing splicing with SSA does not lead to an increase in release times. Conversely, reducing elongation speed with CPT does not slow down splicing. Instead, it leads to an increase in pre-release splicing (*Figure 2E*), as expected if the two processes are independent. Thus, splicing and release can happen in either order, with the order of events determined by kinetic competition. It is this competition which results in stochastic outcomes for the RNA.

What are the physiological implications of this stochastic outcome? Does the cell utilize changes in kinetic balance to alter gene expression? Moreover, is the timing of intron removal of secondary importance to the timing of splicing commitment? While, single-cell variation in alternative splicing has been observed, the mechanism behind this variability has remained elusive (*Waks et al., 2011*; *Lee et al., 2014*). We have found that a single point mutation in U2AF1 that is recurrent in multiple human cancers alters this kinetic balance to favor post-transcriptional splicing of both the β-globin reporter and also endogenous FXR1 mRNAs. Interestingly, single-molecule measurements on fixed cells suggest that post-release splicing may be the preferred pathway for alternatively spliced transcripts (*Vargas et al., 2011*). Our studies on the mutant U2AF1 provide a mechanistic basis for this observation and suggest a role for mutations in the core splicing machinery for the increased levels of 'noisy splicing' which are observed in cancer (*Pickrell et al., 2010*; *Chen et al., 2011*). Furthermore, our time-resolved results indicate that post-release splicing is more efficient than pre-release splicing, which may explain how the timing of intron removal might lead to different outcomes for the message. Such a model relies on a degree of plasticity in spliceosome assembly and function, which has been suggested by in vitro single-molecule measurements (*Hoskins et al., 2011*; *Shcherbakova et al., 2013*). We speculate that the kinetic delay induced by U2AF1-S34F allows either for alternate pairing between 5′ and 3′ss during transcription or for post-release reconfiguration of pre-mRNA which is not yet committed to intron excision in the spliceosome. Other pathological RNA processing defects may also originate from a similar kinetic imbalance. In summary, the single-molecule approach developed here provides a blueprint for dissecting the many competing processes which take place at the earliest stages of gene expression.

## Materials and methods

### Cell line and DNA constructs

The reporter gene vector (*Figure 1—figure supplement 1A*) was constructed with the human β-globin DNA sequence placed under the control of a Tet-responsive promoter, as described previously (*Janicki et al., 2004*; *Darzacq et al., 2007*). Briefly, the 3′ end of the human β-globin sequence was truncated 72 bp upstream of the endogenous stop codon. It was replaced by a cassette coding for the cyan fluorescent protein fused to the peroxisome-targeting sequence serine-lysine-leucine (CFP-SKL),

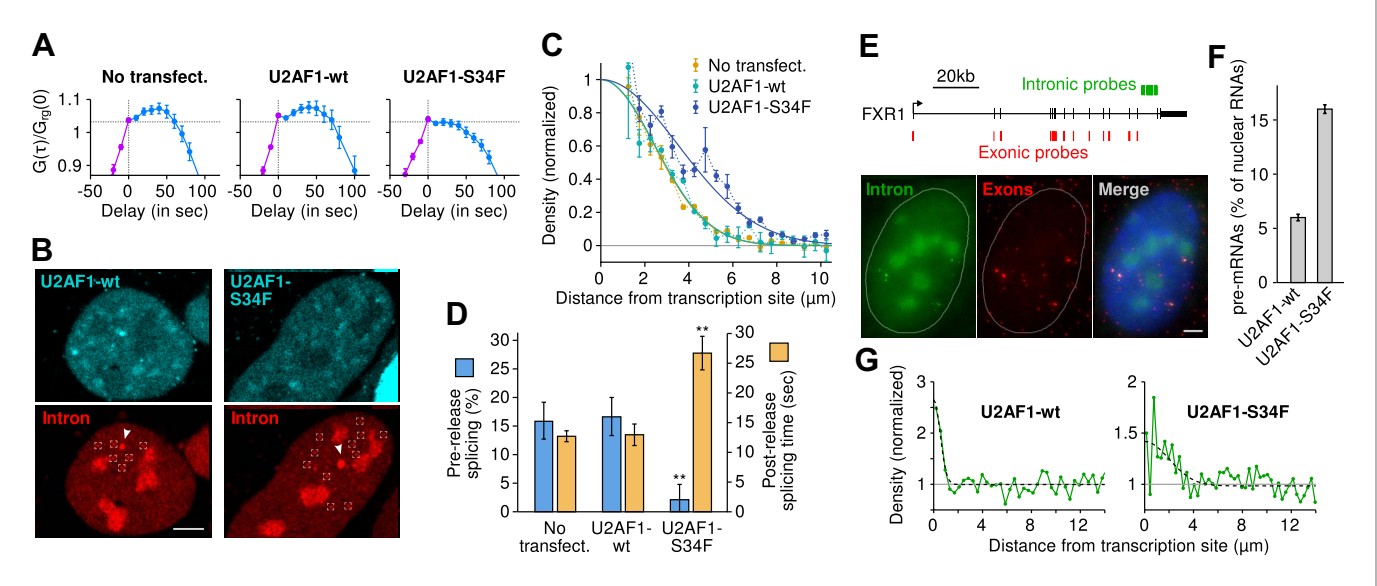

**Figure 6**. The U2AF1-S34F mutant acts as a dominant negative by delaying splicing to post-release. (**A**) Expression of U2AF1-cerulean does not alter pre-release splicing compared to the un-transfected control. Expression of U2AF1-S34F-cerulean abolishes splicing at the TS (horizontal slope of the intron-to-exon cross-correlation, blue curve). (**B**) Pre-mRNA (red, marked by squares) are enriched around the TS (arrows) indicating that splicing still occurs faster than diffusion. The enrichment is broader in the presence of U2AF1-S34F despite the similar spatial distributions of both proteins. (**C**) Gaussian fits onto pre-mRNA radial distance distributions from the TS. (**D**) The U2AF1-S34F mutant defers splicing to occur entirely away from the TS (fractions obtained from model fits in *Table 1*) and increases post-release splicing time. ** p < 0.005 (two-sided z-test *vs* untransfected control). (**E**) Two-color single-molecule FISH on endogenous FXR1 transcripts. Unspliced pre-mRNA (co-localization of intronic and exonic probe) appears in the vicinity of TSs (the 4 bright dual-color spots). (**F**) The fraction of pre-mRNA transcripts in the nucleus in the presence of wt or mutant U2AF1. (**G**) Spatial distribution of pre-mRNAs near TSs in the presence of wt or mutant U2AF1 (N > 400 cells). Radial distributions show density of pre-mRNA normalized by density of mRNA. Bars: 4 μm. Error: SEM over cells (bootstrap).

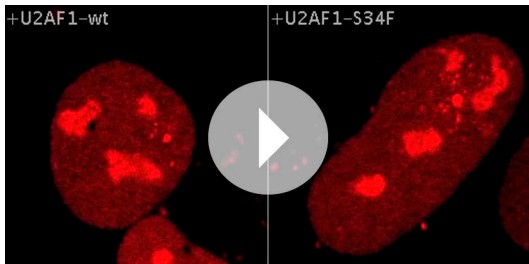

**Video 7**. Spatial distribution of pre-mRNA with wild type or mutant U2AF1. Left and right images show a cell that was transfected with the wild type (wt) or the mutant (S34F) version of splicing factor U2AF1. Both the image show the intron channel. The enrichment of unspliced pre-mRNA (red spots) diffusing in the vicinity of the transcription site is broader in the case of the mutant, showing that splicing rate is slower. Imaging conditions are identical as in **Video 5**.

ending with a translation stop codon. A cassette containing 24 repeats of the MS2 stem loop sequence was inserted in the 3' untranslated region (3'UTR), followed by a bovine growth hormone polyadenylation sequence (BGH-PolyA). In our modified construct, we inserted a cassette containing 24 repeats of the PP7 stem loop sequence into the second and last intron of the gene, approximately halfway between the 5' splice site (5'ss) and the branch point. In order to be the least perturbing, the cassette was inserted 463 bp downstream of the 5'ss and 371 bp upstream of the branch point, and most of the endogenous DNA was conserved (8 bp in the intron were deleted). In addition, we replaced the MS2 cassette with one that is less prone to recombination. The repeating units of the PP7 and MS2 cassettes are composed of two-stem loop blocks, which are then multimerized 12× to get 24× total stem loops:

MS2: GATCCTACGGTACTTATTGCCAAGAAAGCACGAGCATCAGCCGTGCCTCCAGGTCGAATC
TTCAAACGACGACGATCACGCGTCGCTCCAGTATTCCAGGGGTTCATCAG
PP7: CTAAGGTACCTAATTGCCTAGAAAGGAGCAGACGATATGGCGTCGCTCCCTGCAGGTCGA
CTCTAGAAACCAGCAGAGCATATGGGCTCGCTGGCTGCAGTATTCCCGGGGTTCATTAGATC.The
loops themselves are underlined.

The construct was stably integrated into the genomic DNA of U2-OS cells (human osteosarcoma cell line) through transfection of the plasmid followed by a screen for genomic integration. The cell line was made by co-transfection of the reporter plasmid and a puromycin resistance plasmid, followed by selection with puromycin. The cell line also constitutively expresses the PP7 bacteriophage-coat protein fused to the mCherry fluorescent protein (PP7-mCherry) and the MS2 bacteriophage-coat proteins fused to the green fluorescent protein (MS2-GFP). Both are under the control of a ubiquitin promoter and were stably introduced by lentivirus infection as described previously (*Larson et al., 2013*). The clonal cell line used in this study was finally generated by single-cell cloning. The reporter is efficiently expressed and translated as evidenced from imaging of the protein product accumulating in peroxisomes (*Figure 1—figure supplement 1B*).

Cells were grown in DMEM medium (Life technologies, Grand Island, NY) supplemented with 10% fetal bovine serum (FBS, Sigma-Aldrich, St Louis, MO). Cell were induced with 10 or 20 µM doxycycline (Sigma Aldrich, St Louis, MO), at least 24 hr prior to imaging. Imaging was performed in Leibovitz L-15 phenol-free medium (Life technologies, Grand Island, NY) containing the same concentration of doxycycline and FBS. Pharmacological treatments were performed at 48 nM spliceostatin A (Dr Minoru Yoshida, Chemical Genetics Lab, RIKEN Institute, Japan) from 0.5 to 20 hr prior to imaging or 3.75 µM Camptothecin (Sigma, CAS: 7689-03-4) from 1 to 19hr.

U2AF1 (a.k.a. U2AF35) constructs were made from a U2AF35-CFP plasmid provided by Angus Lamond (University of Dundee, UK). The CFP cassette was replaced by a Cerulean one (similar absorption/emission spectra but more photostable). The U2AF1 cassette was then swapped with the mutant version (U2AF1-S34F) provided by Peter Aplan (NCI, NIH).

## Splicing efficiency and poly(A) tail length/site

Cells were transfected with mock or with plasmids expressing either the WT or S34F mutant of U2AF1. Expression of β-globin reporter was induced for 15 hr. Cells were scraped and total RNA isolated using the Qiagen RNA isolation kit. 1 µg of total RNA was used to make first strand using ProtoScript II M-MuLV reverse transcriptase (NEB, Ipsich, MA) and random hexamers (IDT, Coralville, IA) according to the manufacturer's instructions in a final volume of 20 µl. 2 µl of the reverse transcription product was used in a qPCR reaction using IQ Syber Green mix (Bio-Rad, Hercules, CA) in a CFX96 qPCR machine (Bio-Rad, Hercules, CA) according to the manufacturer's instructions. We use varying amounts of G-block DNA (IDT, Coralville, IA) carrying primer pairs generating amplicons of the same size as those being tested. This G-block served as control and was used to generate the standard curves. Primer pairs spanning the junction of Exon2-Intron2, and Intron2-Exon3, and a primer pair amplifying a region of Exon3 served to measure unspliced and total RNA of β-globin reporter respectively.

The unspliced fraction of the reporter RNA was calculated as $10^{\wedge}[\ (Cq_{intron}-K_{intron})/S_{intron} - (Cq_{total}-K_{total})/S_{total}]$ where Cq is the cycle number, and K and S are the constant and slope respectively of the standard curve for the primer pair (*Figure 1—figure supplement 1C*). The splicing fractions calculated using Exon2-Intron2 primers or Intron2-Exon3 primers were similar and were therefore combined together. Error bars are standard errors over four measurements.

U2-OS cells with the integrated β-globin reporter were transfected with mock or with plasmids expressing either the WT or S34F mutant of U2AF1. Expression of β-globin reporter was induced for 15 hr. Cells were scraped and total RNA isolated using the Qiagen RNA isolation kit. 1 µg of total RNA was treated with RNAse H (NEB, Ipswich, MA) in the presence oligo dT according to the manufacturer's instructions. The digested RNA was then phenol:chloroform extracted, ethanol precipitated and suspended in 10 µl of water. A DNA adaptor (5-GGTCACCTTGATCTGAAGC, with a 5-phosphate and 3 amino modification to prevent further ligation) was ligated to 1 µg of either total or RNAse H-treated RNA using T4 RNA ligase 1 (NEB, Ipswich, MA) in a final volume of 20 µl. 10 µl of the adaptor-ligated RNA was then used to synthesize the first strand using ProtoScript II RT (NEB, Ipswich, MA) and the reverse primer 5′-GCTTCAGATCAAGGTGACCTTTTT according to the manufacturer's instructions. The reaction was denatured for 20 min at 80°C and 2.5 µl of the first strand was used as template in a standard PCR reaction using the forward primer 5′-CCAGGGTTCATCAGATCCTATTCTATAGTGTCAC and the reverse primer 5-GCTTCAGATCAAGGTGACCTTTTT. Reaction products were then separated on a 3% agarose gel (*Figure 1—figure supplement 1D*).

## Genomic integration sites and copy number analysis

Whole genome sequencing (paired end sequencing) was used to identify the integration sites of the reporter in the genome. The genomic library was prepared using TruSeq DNA sample preparation

protocol. Samples were sequenced on HiSeq2000 using Illumina TruSeq v3 chemistry. The yield was 389 million reads after filtering. The reads were trimmed to remove low quality sequences (Trimmomatic software) and were aligned to both the human genome (hg19) and the sequence of the β-globin reporter gene plasmid (Bowtie2 and Illumina CASAVA Eland softwares). Paired reads that partially align to both genomic and plasmid sequences were extracted and used to identify possible insertion sites. *Figure 1—figure supplement 2A* shows an example of an insertion site: 4 read pairs align with both with genomic sequence (up to chr8:145,074,337) and plasmid sequence (all from the same position in the plasmid backbone). This defines the 5′ junction of the insertion. Similarly, three read pairs were found partially aligning a few bp downstream (chr8:145,074,349) and with the plasimd up to the beginning of the MS2 cassette, defining the 3′ junction. Multiple copies of the plasmid may have been integrated.

Three insertion sites were identified (*Figure 1—figure supplement 2B*), including two where only one junction was found. All three were confirmed by PCR and the copy number of the reporter in the cell line was obtained as follows. Genomic DNA was isolated from cells (1 × 10 cm plate) by standard protocols, and amplicons were PCR amplified using primer pairs overlapping the putative junctions. The PCR was performed for 24 cycles and the products separated on a 2% agarose gel (*Figure 1—figure supplement 2C*). All four junctions show a band at the expected size. To estimate the copy number of the inserted plasmid, we also use two primer pairs that amplify a region that is internal to the plasmid construct. Each junction should be present in one copy per cell so that, in comparison, the amount of the internal amplicon should reflect the total copy number in the cell line. The gel was quantitated using the ImageJ software (*Figure 1—figure supplement 2D*). To correct for primer efficiency, each primer pair was also used in the same PCR run to amplify varying amount of G-block DNA (IDT, Coralville, IA). The product was quantitated as described above and the resulting calibration curves (*Figure 1—figure supplement 2E*) were used to correct the data. The internal-to-junction ratio of the corrected amounts of PCR products indicates between 4 and 7 total copies of the reporter (*Figure 1—figure supplement 2F*).

## Widefield microscopy and image processing

Microscopy data acquisition from *Figure 1*, *Figure 2* and *Figure 6A,E–G* (and associated supplements) and *Videos 1 to 3* was performed using a custom build wide-field microscope (described in greater details in *Ferguson and Larson, 2013*). It consists in an AxioObserver inverted microscope (Zeiss, Thornwood, NY) with a high aperture objective (Zeiss 63× C-Apochromat) and two Evolve 512 EMCCD cameras (Photometrics, Tucson, AZ). Excitation sources are 488 nm and 594 nm lasers (Excelsior, Spectra Physics, Santa Clara, CA). Typical laser intensities are 250 μW and 50 μW respectively. For imaging Cerulean, we used a 445 LED (Zeiss Colibri) or an X-Cite lamp (Lumen Dynamics). Cells were imaged in 35-mm MatTek dishes (MatTek, Ashland, MA) placed in a Tokai Hit stage incubator (INUB-LPS, Shizuoka-ken, Japan). Average temperature inside the dish was measured at 37°C using a thermocouple. Images were taken every 10 s as z-stacks (7 or 9 images, Δz = 0.5 μm) and in both color simultaneously with two cameras, using an exposure time of 100 ms and for a duration between 45 and 512 frames. Raw images were collected using MicroManager software (*Edelstein et al., 2001*). Maximum intensity projections were computed (e.g. *Video 1*) and used for tracking. *Video 3* was obtained using a shorter exposure time (50 ms) to observe the fluctuations due to diffusing RNAs.

Bicolor fluorescence time traces at the transcription site were generated using a custom software written in IDL (*Source code 1*) that was previously described, with minor modifications to handle bicolor data sets. In each image, diffraction-limited spots are detected using band-pass filtering and refined using an iterative Gaussian mask localization procedure (*Crocker and Grier, 1996*; *Thompson et al., 2002*; *Larson et al., 2005*, *2011*). Trajectories are then generated based on a nearest-neighbor method with a maximal jump distance threshold. If no spot is detected within the threshold distance, the previous location is used as the initial guess for the iterative Gaussian mask localization procedure. Integrated fluorescence intensity over the diffraction-limited spot is collected using a Gaussian mask fit after local background subtraction (*Thompson et al., 2002*). An example of time-lapse video with tracking is shown in *Videos 2*.

Time traces were corrected for photobleaching as follows. In an ideal experiment, the fluorescence intensity histogram of the whole nucleus should stay roughly unchanged throughout the acquisition. We computed smoothed versions (polynomial fit) of the mean and standard deviation (s.d.) of this histogram in each color over time. Time traces from TS tracking were then normalized by the s.d.

(because of the background subtraction in the tracking procedure, only the s.d. of the nucleus histogram acts as a scale factor on a time trace).

Traces were inspected for (i) accurate tracking (portions of inacurate tracking were trimmed off; short traces (<100 frames) were discarded), (ii) in-focus TS (traces where TS reaches the first or last z-plan were discarded), and (iii) signal to noise ratio (highly noisy traces were discarded). Examples are shown on *Figure 1C* and *Figure 2—figure supplement 1A*.

## Fluctuation analysis

For each time trace, autocorrelation and crosscorrelation functions were computed as

$$G_{ab}(\tau) = \langle \delta a(t)\, \delta b(t + \tau) \rangle / \langle a(t) \rangle \langle b(t) \rangle$$

where $\langle \cdot \rangle$ denotes time average, $\delta a(t)$ means $a(t) - \langle a(t) \rangle$, and $a(t)$ and $b(t)$ can be any combination of the red and green time traces $r(t)$ and $g(t)$.

Correlation functions were computed using a multi-tau algorithm (*Wohland et al., 2001*), which iteratively down-samples the signals for increasing time delay. This yields a somewhat uniform spacing of time delay points on a logarithmic scale, reducing the sampling noise at long delays while keeping a high temporal resolution at short delays. When shifting the two signals, non-overlapping ends are not wrapped. See *Figure 2—figure supplement 1A* for examples of correlation functions from single traces.

To reach better statistical convergence, correlation functions from single time traces were averaged together (*Figure 2—figure supplement 1B*). Each point of the single-trace correlation functions was given a weight corresponding to the number of overlapping time points from the signals used in its computation. Bootstrapping was performed to obtain standard error of the mean correlation functions (SEM). Two normalizations were performed prior to bootstrapping: (i) Baseline subtraction: Slow processes (e.g. bursting, cell cycle) may produce a slow decay which adds up to the fast transcription/splicing kinetics. These are usually well separated; see how the decay is much slower after 200–300 s on *Figure 2—figure supplement 1B–F*. The baseline subtraction gets rid of the slow decay, approximating it as a constant offset at short delays. (ii) Normalization with null-delay crosscorrelation $G_{rg}(0)$: All 4 correlation functions were normalized using an estimate of $G_{rg}(0)$. This was done for 2 reasons: (a) doing this normalization prior to bootstrapping promotes a good convergence of the crosscorrelation at short delays 0 (the most informative part of the correlation functions; See *Supplementary file 1*—§2). This constrains the fits to capture precisely the temporal features at short delays. (b) This normalization reduces by 1 the number of free parameters of all the models used to fit the data: On unnormalized correlation functions, changing the initiation rate of transcription simply scales up and down all 4 correlation functions together (See *Supplementary file 1*—§1). Normalizing them gets rid of this degree of freedom. Because most of the time $G_{rg}(0)$ is inaccurate due to shot noise and/or small tracking error (e.g. *Figure 2—figure supplement 1C,E,F*), we use the estimate $(G_{rg}(\Delta t) + G_{gr}(\Delta t))/2$ instead ($\Delta t = 10$ s is the sampling time).

## Confocal microscopy and image processing

We performed imaging with a confocal laser scanning microscope to observe single RNAs diffusing in the nucleus (*Figure 4*, *Figure 6B–C* [associated supplements], *Figure 2—figure supplement 6* and *Videos 5 to 7*). Imaging was performed on a Zeiss 780 confocal microscope with 488/594 nm excitation.

To detect and localize RNAs in the nucleus, we used the same spot-localization algorithm as for widefield microscopy (*Figure 4—figure supplement 1B*). False positive spots due to nucleoli or localized outside of the nucleus were eliminated using a standard masking procedure. Spots were detected independently in both channels (intron and exon) in 9 videos of 50 frames on average taken with a 3.26 s frame interval. Pairs of red and green spots colocalized by less than 250 nm (*Figure 4—figure supplement 1C*) were hence considered single bi-color particles. The TS was tracked and used as a reference for radial distributions of particles (*Figure 4C*, *Figure 4—figure supplement 1D*). These distributions were normalized by the distribution obtained for random locations in the nucleus (i.e., a value of 1 observed indicates a purely uniform distribution). RNAs in the nucleus being relatively sparse, colocalized particles are expected to be mostly true positive, that is unspliced RNAs. On the other hand, a proportion of the red-only and green-only particles may be false positive (e.g., bi-color RNA only detected in one color). mRNAs being much more abundant than pre-mRNA (*Figure 4A*), green-only spots should only have a small portion of false positives. It is however difficult to estimate

what portion of the red-only spots truly represents free lariat and what is a wrong detection/categorization of pre-mRNA. The fact that the red-only radial distributions and the co-localized radial distribution are very similar suggests that red-only spots have a high proportion of false positives (*Figure 4—figure supplement 1D*) and that lariat lifetime is likely shorter than that of post-release pre-mRNA (i.e. < 13 s). Depletion at distances shorter than 1 μm is due to the difficulty for the spot-localization algorithm to locate two spots closer than this distance. The radial distribution of colocalized red–green spots was fit to a Gaussian distribution with three parameters: standard deviation $\sigma$, height $h$, and baseline $y_0$ (*Figure 4—figure supplement 1E*).

We determined the diffusion coefficient of RNAs in the nucleus using raster image correlation spectroscopy (RICS) (*Brown et al., 2008*) (*Figure 4—figure supplement 2*). Imaging was performed in photon counting mode. Time series of 10 frames and 512 × 512 pixels were taken with a 100 μs pixel dwell time, 61 ms line scanning time, and 52 nm pixel size. The $1/e2$ beam waist was determined to be 246 nm in the green channel and 375 nm in the red channel by fitting a 2D Gaussian to a profile through the diffraction limited transcription site. Correlation functions were calculated and fit to a two component diffusion model using the Globals software package developed at the Laboratory for Fluorescence Dynamics at the University of California at Irvine (http://www.lfd.uci.edu/globals/).

Based on the distribution of unspliced transcripts around the TS (i.e., normal distribution with $\sigma$ = 2.421 μm ± 0.087) and a diffusion coefficient of D = 0.12 μm$^2$/s, we deduce that the splicing time following release is on average $\sigma^2/4D$ = 12.71 s ± 0.92. We applied the same analysis to confocal videos obtained on cells transfected with wild-type or mutant version of U2AF1 (*Figure 4—figure supplement 1E*).

We also used confocal images to normalize the intensity of the transcription site by the intensity of single transcripts, allowing us to count the number of nascent transcripts in both channels (*Figure 2—figure supplement 6*). The results were in agreement with the transcription and splicing kinetic parameters we found in our modeling analysis (*Table 1*).

## Fluorescence in situ hybridization (FISH)

To confirm our results obtained on a reporter gene, we performed FISH on the endogenous gene FXR1 (fragile X mental retardation). FXR1 was identified by *Brooks et al. (2014)* as being alternatively spliced in the presence of U2AF1-S34F mutation (i.e., higher retention of the second to last exon). According to our whole genome sequencing data (See above), FXR1 is at a tetraploid locus in our line of U2-OS cells.

We designed 48 probes in the second to last intron (right upstream the alternatively spliced exon) and 48 probes in the exons of FXR1 that are common between all the RefSeq variants, and excluding the two last exons (*Figure 6E*). Probes were generally 20 nucleotides. Intronic and exonic probe sets were synthesized and labeled with cyanine dyes Cy3 and Cy5, respectively, by Biosearch Technologies (Petaluma, CA). U2-OS cells from the same cell line as in the rest of our study were grown on coverslips and transfected with wild type (wt) or mutant (S34F) U2AF1 labelled with Cerulean fluorescent protein, 24 hr prior to fixation. Fixation and hybridization were performed according to the Stellaris RNA FISH protocol (Biosearch Technologies, Petaluma, CA). Coverslips were mounted on microscope slides using mounting media with DAPI (ProLong Gold antifade reagent, Life Technologies).

Imaging was performed on the same widefield microscope as described above. Light sources for imaging DAPI, Cy3, Cy5, and Cerulean were 365 nm, 530 nm, 625 nm, and 445 nm LEDs respectively, from a Zeiss Colibri (Zeiss, Thornwood, NY). The detector was a Hamamatsu ORCA-R2 C10600 camera (Hamamatsu Photonics, Japan). Fields of view were selected for cells with low levels of U2AF1-Cerulean and stacks of nine images with z-step of 0.5 μm were acquired in four colors. Maximum intensity projection images were used for analysis. Diffraction limited spots were identified in the Cy3 and Cy5 channels independently using the same software as above. Nuclear masks were generated from the DAPI channel using CellProfiler software (Broad Institute, http://www.cellprofiler.org). Spots within 200 nm of each other were paired and considered a bicolor particle. Bicolor particles with a fluorescence more than a twofold of that of single RNAs were considered transcription sites (TS).

The measured number of mRNA (exon only particles) was similar with wild-type U2AF1 (29.9 mRNA/nucleus) and with the mutant (35.5 mRNA/nucleus). However, the fraction of pre-mRNAs over total nuclear RNAs was significantly different: 6.0% (±0.3) with the wild type and 16.0% (±0.4) with the mutant (*Figure 6F*). Radial distributions were obtained by computing the distance between single RNAs and all the TSs in a nucleus. Normalized density shown in *Figure 6G* is the density of pre-mRNA

over the density of mRNA. The enrichment at short distances was fit with a Gaussian distribution as for the confocal data. The standard deviation parameter reveals that the local enrichment for pre-mRNA near TSs is broader with U2AF1-S34F (1.83 ± 0.49 μm) than with U2AF1-wt (0.48 ± 0.19 μm). Note that the non-null baseline in the radial distributions also reveals a second population of pre-mRNAs which are spliced slower than diffusion in the nucleus. The proportion of this second population as well as its spatial distribution is not affected by U2AF1-S34F mutation.

## Mechanistic models for RNA synthesis and processing

In *Supplementary file 1*—§2, we showed how different features in the geometry of the correlation functions (e.g., position of specific angles, decay times, change of slope, …) are reflecting different aspects of the transcription/splicing kinetics (e.g., elongation speed, intron and exon dwell times, stochastic co- and post transcriptional splicing, …).

Here, we want to build mathematical models that fully predict the correlation functions given (i) a certain underlying mechanism and (ii) a set of parameters. In essence, we want to ask what molecular mechanisms are consistent with all the above-mentioned kinetic features—and possibly more—that the correlation functions are encoding. The caveat is however that these models necessarily make assumptions on the underlying mechanisms. Hence, we take a general approach in generating a series of simple models (each assuming different mechanisms) and assess which one(s) account for the experimental data.

### Generic approach

To generate a series of 21 minimalistic competing models, we adopt a strategy in two steps. The first step consists in formulating five different general mechanistic schemes, each assuming a different causality relationship between transcription and splicing (*Figure 2—figure supplement 4*):

- •scheme I assumes that splicing is entirely post-release,
- •scheme II assumes no interdependence between splicing and transcription/release,
- •schemes III, IV, and V assume three different checkpoint mechanisms between transcription and splicing (i.e., one process waits for the other).

These schemes are formulated in a generic way by describing the delays between key events in the transcription/splicing process as arbitrary distributions $A(t)$, $B(t)$, $C(t)$, $D(t)$, $E(t)$, and $F(t)$, as shown on *Figure 2—figure supplement 4*. Using this generic description, we are able to derive the general expressions of the correlation functions for each scheme (*Supplementary file 1*—Appendix 2). The second step is to generate multiple simple models from each of these schemes by affecting $A(t)$, $B(t)$, $C(t)$, $D(t)$, $E(t)$, and $F(t)$ with particular distributions. For instance, in a model where everything is deterministic, all the distributions will be Diracs. If a certain step includes a stochastic pause, it will involve an exponential distribution. If a step results from the succession of several biochemical reactions, it will involve a gamma distribution. *Supplementary file 2* summarizes all the different models we derived from each scheme.

For each of these models (described thereafter), changing the parameter value(s) produces different sets of correlation curves. We fit these correlation curves to the experimental data (see examples for the untreated condition on *Figure 2—figure supplement 5A–F*) by fitting all four experimental correlation functions at once. We used a Levenberg-Marquardt non-linear least square fitting procedure, giving twice more weight to the crosscorrelations since they carry more information. Only time points up to typically 350~500 s were used. Fit quality was compared between models using a Bayesian Information Criterion (BIC, See *Supplementary file 1*—Appendix 3) on *Figure 2—figure supplement 5* (the lower the BIC, the better the fit). The BIC favors simpler models: if two models with different number of parameters fit equally well the data, the BIC penalizes the one with extra degrees of freedom so that only the simplest model is retained.

In addition, as indicated in *Supplementary file 2*, each model may or may not be consistent with:

- •the geometrical properties of $G_{rg}(0)$ described in the previous section (*Figure 2—figure supplement 2C–D*; example, scheme I necessarily yields a flat $G_{rg}(0^+)$ by construction), or
- •the fact that transcripts can be released unspliced from the transcription site (*Figure 4A–B*).

### Details of schemes and models

Schematics of each scheme are shown on *Figure 2—figure supplement 4* and the models generated from them are listed in *Supplementary file 2*. The general forms of the correlation functions are derived in *Supplementary file 1*—Appendix 2.

## Scheme I—splicing is post-release

Here, the red signal always remains as long as the RNA is being transcribed and drops coincidentally with the green signal, when the transcript is released. Model I.1 is the simplest possible: it assumes that everything is deterministic, that elongation proceeds with a constant speed, and that transcript release is instantaneous. It has a single parameter v, representing the speed of the polymerase. In models I.2, we add an elongation pause between the two cassettes. This can represent a pause anywhere between the cassettes, such as, in particular, right after the 3′ splice site (3′ss). This model has two parameters: the elongation speed v and the mean pause time ps. Model I.3 is constructed similarly by introducing an exponentially distributed pause at the 3′ end of the gene. Model I.4 includes pauses both after the 3′ss and at the 3′ end of the gene.

## Scheme II—no crosstalk between transcription and splicing

Once the 3′ss is passed, splicing and transcription proceed independently and may complete in either order for different transcripts as a result of their own stochastic kinetics. The red signal drops either when splicing occurs (after $F(t)$) or when the transcript is released (after $E(t)$), whichever of the two happens first. Models II.1 and II.2 consider a deterministic elongation (equivalent of model I.1) with either a deterministic or a stochastic splicing time. Models II.3 to II.5 include a pause at the 3′ss, at the 3′ end of the gene, or at both (as for models I.2 to I.4), but also include here a deterministic splicing time. Of all the models described so far, the experimental data is best fit by model II.4 where splicing is deterministic and 3′ end dwell time exponential. Hence, we generated three other closely related models: II.6 where splicing is exponential and 3′ end dwell time deterministic, II.7 where both are exponential, and II.8 where 3′ end dwell time is exponential and splicing follows a gamma distribution.

## Scheme III—checkpoint at 3′ splice site: Elongation pauses after the 3′ss until intron is spliced

This is an obligatory checkpoint because splicing has to complete before the polymerase keeps elongating, as opposed to models II.3 and II.5 where there is a pause after the 3′ss regardless of the splicing kinetics. By construction, the red signal always drops before the green signal starts rising. Hence, $G_{rg}(0)$ is necessarily null and flat on both sides of the y-axis (first case of *Figure 2—figure supplement 2C*), making it obviously unable to fit our experimental data. For the sake of illustration, we derived two models, respectively with an exponential and a deterministic splicing/pausing time. Both fit very badly the experimental data (*Figure 2—figure supplement 5D*).

## Scheme IV—checkpoint at exon end/termination site

Splicing can only take place after the polymerase has reached the termination site (or the 3′ end of the last exon). Once splicing happens, release of the transcript may require an extra time (distribution $F(t)$). In models IV.1 to IV.3, when the polymerase reaches the termination site, splicing occurs either immediately (IV.1), or after an exponentially (IV.2) or gamma distributed (IV.3) delay. Then, the RNA is released after an exponentially distributed time (e.g., for additional 3′ end processing). These three cases correspond to a checkpoint mechanism that holds the splicing process respectively at its last biochemical step, or at 1 or several steps prior to its completion. In this scheme, RNAs cannot be released unspliced. So the only possibility for $G_{rg}(0)$ to show both a break of slope and a rising slope at $0^+$ would be that splicing and release occur simultaneously for a fraction of the transcripts. Hence, we built model IV.4 which is similar to model IV.2 except that, for a fraction of the transcripts, release is immediate after splicing. This corresponds to a mechanism where both splicing and RNA 3′-end processing take (independent) exponential times after the termination site is reached, but the RNA is only released when both processes have come to completion.

## Scheme V—checkpoint for release

After the polymerase has passed the 3′ss, elongation and splicing happen independently of each other as in scheme II. However, here, the transcript can only be released after the intron is removed. In models V.1 to V.3, the splicing time is respectively deterministic, exponentially distributed, or follows a gamma distribution, corresponding to different number of biochemical reaction in the splicing process.

## Model comparison

Evaluating all the models on each experimental data set (*Figure 2—figure supplement 5*) using the Bayesian Information Criterion (BIC, *Supplementary file 1*—Appendix 3), we find that, the most likely models are either II.4 or IV.4 in the control case, as well as in many of the other cases. Models IV.3 and V.3 perform well in a number of cases, but only V.3 is consistent with all the geometric features of $G_{rg}(0)$ (*Supplementary file 2*), making it also a good candidate to explain the data. Complementary to the BIC, a visual inspection of the fits (*Figure 2—figure supplement 5A–F*) also illustrates how the different models perform. In particular, how badly models II.1 to II.3 fit the data indicates the need for a pause at 3′ end.

To conclude our modeling and data fitting analysis, three candidate models are able to reproduce the data accurately: II.4, IV.4, and V.3. All three models agree on the following:

- the delay between splicing and release is stochastic,
- only a fraction of the RNAs are spliced strictly before release,
- pre-release splicing occurs mostly (or exclusively) at the 3′ end of the gene,
- all three models give similar values for delays between events at the transcription site (so that any may be used to report these delays).

These models differ on the causality between events: whether splicing and transcription are independent or one waits for the other cannot be known purely from the correlation curves themselves. Model II.4, however, appears as the simplest of all three since it assumes independence instead of a checkpoint mechanism and has only three parameters. In addition, several strong lines of evidence point toward model II.4. Indeed, only this model is consistent with the following observations:

- inhibiting splicing does not affect 3′-end dwell time (*Table 1*),
- inhibiting elongation does not affect splicing time (*Table 1*),
- some unspliced RNAs can be released from the transcription site (*Figure 4* and *Video 5*).

These observations argue that splicing and transcription are independent. Hence, we conclude that model II.4 is the most likely mechanism and use it in the main article to report parameter values.

## Error on model parameters and statistical tests

We used error propagation from the correlation functions to obtain SEM on the fitting parameters. To assess statistical significance in *Table 1*, a standard two-sided z-test was systematically performed, for each model parameter, between the untreated condition and all the other conditions.

# Acknowledgements

The authors would like to thank Tom Johnson and Joseph Rodriguez for assistance with qRT-PCR and Nico Sturman and Arthur Edelstein for help with µManager. Spliceostatin A was a kind gift from M. Yoshida, RIKEN ASI. U2AF1 (wt and S34F) were provided by Angus Lamond and Peter Aplan. Aaron Hoskins and Joseph Rodriguez provided critical feedback on the manuscript. The cell line was sequenced by the sequencing facility of NCI, Frederick. The authors would like to acknowledge initial support from R01GM086217 to Robert Singer. The authors thank members of the Janelia Farm/HHMI Transcription Imaging Consortium for helpful discussions.

# Additional information

### Funding

| Funder | Grant reference number | Author |
|---|---|---|
| National Cancer Institute | 1ZIABC011383-03 | Matthew L Ferguson, Murali Palangat, Daniel R Larson |
| National Institute of Diabetes and Digestive and Kidney Diseases | | Antoine Coulon, Carson C Chow |

The funders had no role in study design, data collection and interpretation, or the decision to submit the work for publication.

## Author contributions

AC, MLF, Conception and design, Acquisition of data, Analysis and interpretation of data, Drafting or revising the article; VT, Conception and design, Contributed unpublished essential data or reagents; MP, Acquisition of data, Analysis and interpretation of data, Drafting or revising the article; CCC, Analysis and interpretation of data, Drafting or revising the article; DRL, Conception and design, Analysis and interpretation of data, Drafting or revising the article

## Author ORCIDs

Matthew L Ferguson, http://orcid.org/0000-0003-0760-757X
Valeria de Turris, http://orcid.org/0000-0003-0872-185X
Daniel R Larson, http://orcid.org/0000-0001-9253-3055

## Additional files

### Supplementary files

• Supplementary file 1. Mathematical analysis of the correlation functions.

• Supplementary file 2. Models from mechanistic schemes. From each scheme depicted in *Figure 2—figure supplement 4*, we derive a series of models by simply affecting the arbitrary distributions $A(t)$, $B(t)$, $C(t)$, … to specific ones (columns *Time distributions*). Each model represents a specific mechanism (column *Description*) and involves a different number of parameters (column *Params*). Some models may be excluded simply because, by construction, they cannot reproduce certain basic geometric properties of the crosscorrelation functions (columns Features of $G_{rg}(0)$; properties described in *Figure 2—figure supplement 2C*), or because they do not allow for unspliced transcripts to be released (last column), as observed experimentally (*Figure 4*, *Figure 4—figure supplement 1* and *Video 5*). 3'ss: 3' splice site.

• Source code 1. Source code and executable file for the spot tracking software.

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
