## [Decision Letter]

Thank you for sending your work entitled “Kinetic competition during the transcription cycle results in stochastic RNA processing” for consideration at *eLife.* Your article has been favorably evaluated by James Manley (Senior editor), a Reviewing editor, and 3 reviewers.

The following individuals responsible for the peer review of your submission have agreed to reveal their identity: Douglas Black (Reviewing editor); Alvaro Sanchez and Xavier Darzacq (peer reviewers). A further reviewer prefers to remain anonymous.

The Reviewing editor and the other reviewers discussed their comments before we reached this decision, and the Reviewing editor has assembled the following comments to help you prepare a revised submission.

This paper from Larson and colleagues examines the relative kinetics of pre-mRNA synthesis, splicing, and transcript release from the DNA template. This is an important issue for understanding the coupling of these reactions and an area of substantial interest in the field. The authors use single molecule imaging of a tagged beta-globin RNA substrate carrying fluorescent labels in an intron and the 3' exon to follow the appearance and disappearance of the two labels in living cells. They develop a new and elegant analysis that applies correlation spectroscopy to the intron and exon signals to extract reaction rates for transcriptional elongation, splicing, and release from the gene locus. As seen by others, they find that some splicing occurs at the locus and some after release. Using inhibitors to alter the rates of either transcription or splicing, they show that these rates are essentially independent of each other. They find that there is a significant delay between the time of transcript completion and its release from the locus, with the fraction of splicing seen at the locus in kinetic competition with the elongation and release rates. The intron containing transcripts that are released from the locus appear to be very rapidly spliced as measured in relation to their diffusion from the site of transcription. Finally, the authors present interesting experiments using their system to analyze the effects of a dominant mutation in the U2AF1 splicing factor. U2AF1 mutations have recently been implicated in myelodysplastic syndromes, and the authors show that expression of the mutant factor significantly delays splicing to be mostly after release from the locus.

The authors present a new analysis of an important problem, and the data in this paper are very strong. All of the reviewers felt that this was an important contribution. However, all the reviewers also found multiple problems with the description of the experiments, the interpretation of the data, and other issues with the language. These are summarized as follows.

1) The use of the term “stochastic” in the Title and elsewhere and the comparison of stochastic transcription initiation to “stochastic RNA processing” was confusing. The splicing was not necessarily initiating randomly and was completing at a definable rate. Describing the process as a kinetic competition was more apt, but this should be more clearly defined earlier in the paper as a competition between the splicing rate and the combined elongation and release rates.

2) Throughout the paper, splicing occurring at the gene locus is described as cotranscriptional. Although this usage is seen in other papers, it is not strictly correct. Brody et al and others have described polyA RNA at the locus and the authors here describe a significant dwell time after the 3' end is synthesized. Thus, at least some of the splicing at the locus is presumably taking place on transcriptionally complete RNAs. Making this clearer would avoid confusing constructions like: “co-transcriptional splicing is rarely completed during elongation”. Similarly, the authors seem to imply in places that 3' cleavage is occurring at the time of release. This is not shown and there is data from others to indicate it may at the beginning of the dwell time not the end. Since single molecule transcription experiments in vitro do not show delayed release, the authors might discuss what the delayed release in vivo might mean. Paused polymerase prior to the poly-A site, the need to pass some stage in maturation prior to release, a process that anchors material close to a locus that is absent from an in vitro system, or some other possibility?

3) The statements that post-transcriptional splicing is more efficient or faster than cotranscriptional splicing (or splicing pre and post release) are also not supported. It is possible that released transcripts have all past a rate limiting step in spliceosome assembly allowing the catalytic step to occur very soon after release. This also weakens their argument that the data indicate a lack of a checkpoint prior to release. While it seems clear that unspliced RNA can be released, this release of RNA may require completing a particular stage of assembly.

4) There are a number of aspects of the reporter system and its validation that are not described. Some of this may already be published but it should be mentioned and referenced. The constructs to build the gene arrays are poorly described and the origin/construction of the globin gene. The authors should also mention how the kinetics of binding of the MS2 and PP7 fusion proteins to RNA affect interpretation of the results. The authors do not seem to see well defined red spots corresponding to the spliced intron, only spliced exons (green) or unspliced pre-RNA (red+green). Is the half-life of the excised intron known? Could at least the PP7-bound portion of the intron be protected by the coat proteins and thus more stable? The data would argue not, but has this been looked at? Could the coincident red and green spots seen to move away from the locus actually be an excised intron still in complex with the ligated exons? It should also be noted that any synthetic reporter may be missing features found in an endogenous gene and may yield different results. This does not take away from the data, but it is a limitation of the experimental system.

5) The authors describe testing 21 models to find the best fit to the correlation data. This is very hard to follow. It would be helpful to start with the 5 different mechanistic scenarios and explain how specific model solutions exclude them. Showing a few examples of how a particular hypothesis when translated into a modeled simulation is not compatible with the data might also help the reader. In the main text it is important to give some idea of which conclusions are clearly ruled out from the modeling and which are simply disfavored because their fit to a model is slightly less than others. This is not easy material to explain to a general audience but some sense of how the simulations led to the conclusions is important to convey.

6) More discussion of how these results fit with others in the field is needed. Recent papers from Carmo Fonseca (45), Huranova et al, Brody et al and others have also used imaging methods to examine transcription and splicing kinetics, sometimes reaching different conclusions. For non-specialists in the audience, it would be very helpful to have some description of how the methods and results compare, and what might lead to different conclusions.

---

## [Author Response]

*1) The use of the term “stochastic” in the Title and elsewhere and the comparison of stochastic transcription initiation to “stochastic RNA processing” was confusing. The splicing was not necessarily initiating randomly and was completing at a definable rate. Describing the process as a kinetic competition was more apt, but this should be more clearly defined earlier in the paper as a competition between the splicing rate and the combined elongation and release rates*.

We clarified this point in the text by defining what we mean by “stochastic RNA processing” and how it is comparable to stochastic transcription. In stochastic transcription, events such as a gene switching ON and OFF are stochastic but still have a definable rate. The non-deterministic outcome is the amount of a given RNA in a cell over time. Similarly, in RNA synthesis and processing, events are stochastic and also occur at a definable rate. The non-deterministic outcome is the pathway taken to go from nascent pre-mRNA to mRNA. The simple fact that our data cannot be described by a deterministic model indicates that, by definition, RNA processing is stochastic.

2) Throughout the paper, splicing occurring at the gene locus is described as cotranscriptional. Although this usage is seen in other papers, it is not strictly correct. Brody et al and others have described polyA RNA at the locus and the authors here describe a significant dwell time after the 3' end is synthesized. Thus, at least some of the splicing at the locus is presumably taking place on transcriptionally complete RNAs. Making this clearer would avoid confusing constructions like: “co-transcriptional splicing is rarely completed during elongation”. Similarly, the authors seem to imply in places that 3' cleavage is occurring at the time of release. This is not shown and there is data from others to indicate it may at the beginning of the dwell time not the end. Since single molecule transcription experiments in vitro do not show delayed release, the authors might discuss what the delayed release in vivo might mean. Paused polymerase prior to the poly-A site, the need to pass some stage in maturation prior to release, a process that anchors material close to a locus that is absent from an in vitro system, or some other possibility?

The reviewers raise a valid point. Part of the confusion arises from the terminology that is already established in the literature, which is admittedly vague. Our data indicates that RNA is retained for ∼2 minutes after synthesis should be complete. We cannot differentiate between a pause in elongation at the 3’ end and retention of cleaved RNA by tethering to RNAPII or chromatin. By extension, we also cannot determine whether these retained RNAs are poly-adenylated. We have clarified this interpretation by considering every occurrence of the words ‘co-transcriptional’ and ‘post-transcriptional’ throughout the manuscript and either replacing the terms with ‘pre-release’ and ‘post-release’ or reformulating the description all together. We have also made the distinction between release and cleavage.

We also point out that the alternative explanations for the 3’ pausing enumerated by the referees were present in the original draft.

*3) The statements that post-transcriptional splicing is more efficient or faster than cotranscriptional splicing (or splicing pre and post release) are also not supported. It is possible that released transcripts have all past a rate limiting step in spliceosome assembly allowing the catalytic step to occur very soon after release. This also weakens their argument that the data indicate a lack of a checkpoint prior to release. While it seems clear that unspliced RNA can be released, this release of RNA may require completing a particular stage of assembly*.

We disagree with this interpretation of our data and have made changes to the text to clarify our results. As now explained, we do think our data shows that splicing is faster post-release than at the site of transcription. If the splicing kinetics were unaffected by the departure of the RNA from the transcription site, one would expect a post-release splicing time of 137 s on average. (This time corresponds to the 267 s for the total splicing time, minus the elongation and release times, calculated only for the transcripts that are released unspliced). The fact that we see a much shorter time (13 s) indicates that the pre-release splicing rate cannot be the same as the post-release splicing rate. In fact, the post-release kinetics are unexpectedly fast. The argument that a transcript is probably highly committed before release, with most of the spliceosome assembled, is a persuasive one with which we find ourselves in agreement, but this statement is not the same as a checkpoint requirement. Our splicing inhibition experiment would appear to be clear-cut in this case: treatment with spliceostatin A, which inhibits U2 snRNP assembly, has no measurable effect on the release time. So if there is a checkpoint, it is before U2 recruitment.

*4) There are a number of aspects of the reporter system and its validation that are not described. Some of this may already be published but it should be mentioned and referenced. The constructs to build the gene arrays are poorly described and the origin/construction of the globin gene*.

We have added more information about the reporter construct, the coat proteins, and the cell lines to the Methods.

*The authors should also mention how the kinetics of binding of the MS2 and PP7 fusion proteins to RNA affect interpretation of the results*.

The association rate of the coat protein has a negligible effect on our kinetic measurements. The Greenleaf lab has measured the association rate of MS2 for >3,000 hairpin variants (13). The variant we use has a k_on_ of 0.000538011 nM^-1^s^-1^. For 1 μM coat protein levels, the average time to associate is 1.85 s. The average time to synthesize the 60 bp repeating unit of the stem loop cassette at 2.6 kb/min is 1.4 s. Therefore, on average, the previous stem loop has already bound coat protein before synthesis of the next stem loop is completed. More importantly, this association time is two orders of magnitude lower than the kinetic quantities we report in this manuscript. We have added a footnote to this effect in the Results.

The authors do not seem to see well defined red spots corresponding to the spliced intron, only spliced exons (green) or unspliced pre-RNA (red+green). Is the half-life of the excised intron known? Could at least the PP7-bound portion of the intron be protected by the coat proteins and thus more stable? The data would argue not, but has this been looked at?

The question of imaging lariats is a difficult one. In our case, we do detect a certain amount of red-only particles, and we reported these results in our original submission (see Figure 4—figure supplement 1). However, we cannot say with confidence whether these are actual red-only particles (i.e. free lariats) or simply mis-classified pre-mRNA. If we take this number at face-value, the red-only fraction of the total nuclear RNA pool is ∼ 2%. Based on the spatial distribution around the TS, the lariat lifetime < 10 s. We clarify this aspect of our assay in the main text and explain in greater detail in the Methods section. We also explain why we have much better confidence in the green-only and the dual-color particles.

*Could the coincident red and green spots seen to move away from the locus actually be an excised intron still in complex with the ligated exons? It should also be noted that any synthetic reporter may be missing features found in an endogenous gene and may yield different results. This does not take away from the data, but it is a limitation of the experimental system*.

We now address these two caveats brought up by the reviewers: that dual-color particle potentially being lariat and mRNA still in complex and that a reporter gene may behave differently from an endogenous one.

*5) The authors describe testing 21 models to find the best fit to the correlation data. This is very hard to follow. It would be helpful to start with the 5 different mechanistic scenarios and explain how specific model solutions exclude them. Showing a few examples of how a particular hypothesis when translated into a modeled simulation is not compatible with the data might also help the reader. In the main text it is important to give some idea of which conclusions are clearly ruled out from the modeling and which are simply disfavored because their fit to a model is slightly less than others. This is not easy material to explain to a general audience but some sense of how the simulations led to the conclusions is important to convey*.

Following the reviewers’ advice, we reformulated the description of the modeling and model testing. This is now hopefully more understandable by the non-specialist readers. We first describe the 5 different schemes, then we explain that two of them can be ruled out even without having to fit the data and we decide between the other 3 by fitting the correlation functions.

*6) More discussion of how these results fit with others in the field is needed. Recent papers from Carmo Fonseca (*[45]*), Huranova et al, Brody et al and others have also used imaging methods to examine transcription and splicing kinetics, sometimes reaching different conclusions. For non-specialists in the audience, it would be very helpful to have some description of how the methods and results compare, and what might lead to different conclusions*.

We have expanded the discussion of how our results compare to other kinetic measurements in the literature.

Buenrostro, J.D., Araya, C.L., Chircus, L.M., Layton, C.J., Chang, H.Y., Snyder, M.P., and Greenleaf, W.J. (2014). Quantitative analysis of RNA-protein interactions on a massively parallel array reveals biophysical and evolutionary landscapes. Nat Biotech 32, 562-568.